# Portrait of Molecular Signaling and Putative Therapeutic Targets in Prostate Cancer with *ETV4* Fusion

**DOI:** 10.3390/biomedicines10102650

**Published:** 2022-10-20

**Authors:** Ye Ji Shin, Jae Won Yun, Hong Sook Kim

**Affiliations:** 1Department of Biological Sciences, Sungkyunkwan University, Suwon 19419, Korea; 2Veterans Medical Research Institute, Veterans Health Service Medical Center, Seoul 05368, Korea

**Keywords:** cellular signaling pathways, drug repurposing, *ETV4*, prostate cancer

## Abstract

Gene fusion between androgen receptor (AR) response genes and E26 transformation-specific (ETS) family members increases the gene expression of ETS family members, and promotes tumorigenesis in prostate cancer. However, the molecular features of *ETV4* fusion in prostate cancer are not fully understood, and drugs targeting *ETV4* fusion have not been developed. To examine key cellular signaling pathways and explore therapeutic targets and drugs for *ETV4*-fusion-positive prostate cancer, we analyzed RNA sequencing data and clinical information for prostate cancer. The *ETV4*-fusion-positive group was selected through prior study and analysis comparing *ETV4*-fusion-positive and -negative groups was conducted using a Pearson correlation test. We obtained 393 genes correlated with *ETV4* expression. Pathway analysis was performed using over-representation analysis (ORA), and six cancer-specific molecular signaling pathways (the irinotecan pathway, metabolism, androgen receptor signaling, interferon signaling, MAPK/NF-kB signaling, and the tamoxifen pathway) were altered in the *ETV4*-fusion-positive group. Furthermore, a gene–drug database was used to find an actionable drug and therapeutic target for the *ETV4*-fusion-positive group. Here, we have identified significantly altered genes and oncogenic signaling pathways in *ETV4*-fusion-positive prostate cancer, and we suggest therapeutic targets and potential drugs for *ETV4*-fusion-positive prostate patients.

## 1. Introduction

Prostate cancer is the second-most frequent cancer type in males following lung cancer, and it ranked as the fifth-leading cause of death in males in 2020 [1]. It is also the most commonly diagnosed type of cancer in developed countries, such as those in North America and Northern/Western Europe.

Prostate cancer frequently harbors the chromosomal rearrangement of the E26 transformation-specific (ETS) gene, and the most frequent fusion partner gene is *TMPRSS2*. 

*ERG* and *TMPRSS2* fusion is found in about 50% of prostate cancers [2], and *ETV1* and *TMPRSS2* fusion is found in 5% of prostate cancers. Fusion of the ETS gene results in the high expression of *ERG* or *ETV1*, and it is an important cause of prostate tumorigenesis and tumor development [3,4]. In addition to its oncogenic effects, the clinical relevance of ETS gene fusion has been well described [4,5]. Since ETS gene fusion was first discovered, novel 5′ and 3′ partner genes, such as *ETV4*, *ETV5*, and *SKIL* have been discovered in studies [5]. The structural variation of ETS variant transcription factor 4 (*ETV4*), defined as another driver mutation, was observed in 2% of prostate cancers [6]. *TMPRSS2* is a major fusion companion of *ETV4*, and a significant portion fuses by gene-intergenic fusion, an unusual mechanism which produces chimeric RNA [7]. Other genes, such as *STAT3* and *PMEPA1*, are also known to fuse with *ETV4*. Also, there are some cases where the mechanism by which *ETV4* is highly expressed in prostate cancer is not well-defined. The overexpression of *ETV4* promotes cell proliferation by upregulating cancer-related genes and the epithelial–mesenchymal transition (EMT) by activating EMT-specific transcription factors [8]. *ETV4* increases metastasis by activating PI3K kinase-RAS signaling in mouse prostate cancer models [9]. Overall, the oncogenic effects of *ETV4* are well-described in both in vivo and in vitro models [8]. However, molecular features, such as gene expression and signaling pathways, in *ETV4*-fusion-positive prostate cancers have not been fully elucidated. Indeed, therapeutic targets and candidate targeted drugs for *ETV4*-overexpressing prostate cancer remain unknown. Thus, we here systematically analyzed RNA sequencing (RNA-seq) data obtained from The Cancer Genome Atlas (TCGA) database and identified oncogenic signaling pathways activated in *ETV4*-fusion-positive prostate cancer, which results in *ETV4* overexpression. We further performed gene–drug network analysis and investigated therapeutic targets and candidate drugs in *ETV4*-fusion-positive prostate cancer.

## 2. Materials and Methods

### 2.1. Selection of Case and Control Groups for Analysis

Structural variation, mutation information, and gene expressions from The Cancer Genome Atlas (TCGA) prostate cancer molecular taxonomy study was used to clarify subtypes of prostate cancer [10]. A total of 14 *ETV4*-fusion-positive samples were selected as the case group, and 86 samples without any driver mutations, such as *ERG* fusion, *ETV1* fusion, *SPOP* mutation, *FOXA1* mutation, or *IDH1* mutation, were selected as the control group.

### 2.2. Selection of Genes Associated with ETV4 Expression

RNA expression data with upper quantile normalization was obtained from the TCGA Prostate Adenocarcinoma (PRAD) study from the Broad GDAC Firehose website (http://gdac.broadinstitute.org, accessed on 12 December 2021). Sample type code 01, indicating primary solid tumor samples, and sample type code 11, indicating adjacent normal tissue, were used in this study. *ETV4* fusion with androgen-responsive genes results in *ETV4* overexpression in prostate cancers [6]; thus, we selected genes associated with *ETV4* expression via the Pearson correlation test using an absolute value of Pearson correlation coefficient over 0.3 (|R| > 0.3), assuming that the expression fluctuation of downstream genes affected by *ETV4* would be proportional to the expression of *ETV4*. 

The CancerMine database (http://bionlp.bcgsc.ca/cancermine, accessed on 12 December 2021) was used to identify cancer-related genes [11]. This database contains genes and their functional effects, such as tumor suppressor, oncogene, and driver, with cancer types.

### 2.3. Pathway Analysis

The genes significantly correlated with *ETV4* expression in prostate cancer were used to perform over-representation analysis (ORA) via ConsensusPathDB (CPDB, http://cpdb.molgen.mpg.de, accessed on 22 June 2021) with the options of allowing a minimum of two genes to overlap from the input list and a *p*-value < 0.01 cutoff. Biocarta, Ehmn, Humancyc, Inoh, Kegg, Manual upload, Netpath, Pharmgkb, Pid, Reactome, Signalink, Smpdb, and Wikipathways, which were curated and provided by CPDB, were used for pathway analysis. 

### 2.4. Therapeutic Targets and Candidate Drugs 

A drug-target database containing target genes, variants, disease, and drugs was obtained from the Clinical Interpretation of Variants in Cancer platform (CIViC, https://civicdb.org/home, accessed on 22 June 2021). Gene–drug network analysis was performed using genes involved in cancer-specific pathways of *ETV4*-fusion-positive prostate cancer, and therapeutic targets and potential actionable drugs were obtained via the network analysis.

### 2.5. Statistical Analysis and Visualization

Clinical characteristics, such as age, prostate-specific antigen (PSA) level, Gleason score, tumor stage (T stage and M stage), laterality, and ethnicity were analyzed between the *ETV4*-fusion-positive and -negative groups to check differences in clinical information via the moonBook package (https://cran.r-project.org/web/packages/moonBook/, accessed on 7 October 2021) in software R version 3.6.3. Because the clinical information provided from TCGA is limited, only the available data sets were used for this statistical analysis. 

Genes significantly correlated with *ETV4* expression were identified via a Pearson correlation test using the cor.test function in R. Cancer-specific cellular signaling pathways and involved genes altered in *ETV4*-fusion-positive prostate cancer were visualized at the gene level using an R package named ComplexHeatmap [12]. Graphs, except for heatmaps, were visualized using the R package ggplot2, and statistical significance was calculated and labeled by the R package ggpubr. The gene–drug network and gene-pathway network were visualized via Cytoscape version 3.8.2 [13].

## 3. Results

### 3.1. Comparison of Patient Characteristics

*ETV4*-fusion-positive prostate cancers were selected based on structural variation data and gene expression data according to a previous report [10]. Patient characteristics between 14 *ETV4*-fusion-positive prostate cancers and 86 *ETV4*-fusion-negative prostate cancers were analyzed. Age, PSA level, Gleason score, tumor stage (T stage and M stage), laterality, and ethnicity were compared between the *ETV4*-fusion-positive and -negative prostate cancer groups, and no significant difference was observed (Table 1). 

### 3.2. Genes Associated in ETV4-Fusion-Positive Prostate Cancer

The mRNA expression of *ETV4* was higher in *ETV4*-fusion-positive prostate cancer, as compared to that in *ETV4*-fusion-negative prostate cancer. A Pearson correlation test using an absolute correlation coefficient value over 0.3 (|R| > 0.3) was applied to identify genes significantly correlated with *ETV4* expression. A total of 393 genes were obtained, and interestingly, except for 10 genes, most were positively correlated (Appendix A). These 393 genes were further analyzed using cancer-related genes from the CancerMine database, and 122 genes were defined as cancer-related genes (Appendix A). Of these, 19 genes had previously reported oncogenic functions in prostate cancer (Appendix A), which suggests that these genes could specifically function in *ETV4*-fusion-positive prostate cancers. Furthermore, the oncogenic role of approximately 100 of the cancer-related genes in *ETV4*-fusion-positive prostate cancers were suggested, and consistently, as an example, a recent study reported that *CDK19* is highly expressed in prostate cancer and increases aggressiveness and regulates the progression of prostate cancer [14]. *CDK19* was the cancer-related gene most highly correlated with *ETV4* expression (Appendix A), confirming its oncogenic function in *ETV4*-fusion-positive prostate cancer. 

### 3.3. Cellular Signaling Pathways Associated with ETV4-Fusion-Positive Prostate Cancer

Cellular signaling pathway analysis was performed using the selected 393 genes via ORA to investigate altered signaling pathways in the *ETV4*-fusion-positive prostate cancer group. Six cancer-specific signaling pathways—The irinotecan pathway, metabolism, androgen receptor signaling, interferon signaling, MAPK/NF-kB signaling, and the tamoxifen pathway—were identified, and 86 genes were involved in these cancer-specific pathways (Figure 1). We further dissected the metabolic pathways and identified lipid metabolism, carbohydrate metabolism, and biosynthesis of cofactors, which were targeted by at least five genes (Appendix A).

A total of 85 genes were significantly highly expressed, and only *PXN* was downregulated in *ETV4*-fusion-positive prostate cancer (Appendix A). Paxillin, encoded by the *PXN* gene, is an oncogenic protein. Paxillin relates to cancer proliferation and metastasis, and *PXN* gene expression is relatively high in many cancers, including prostate cancer [15,16,17]. *PXN* expression was relatively high in *ERG1*- and *FLI1*-fusion-positive prostate cancers and *SPOP*-mutant prostate cancer, but it was relatively low in *ETV1*- and *ETV4*-fusion-positive prostate cancers (Appendix A). Consistently, a positive correlation between *PXN* and *ERG* (R = 0.5) and a negative correlation between *ETV1*/*ETV4* and *PXN* (R = −0.3) were observed. Moreover, *PXN* expression in *ETV4*-fusion-positive prostate cancer was lower compared to normal (Appendix A). 

Gene-pathway network analysis was further performed (Figure 2). Six genes, *CYP1B1*, *CYP3A4*, *HSPA5*, *RAN*, *SULT2A1*, and *TOP1*, were involved in more than two pathways, and multiple targeted pathways included the tamoxifen pathway, metabolism, the irinotecan pathway, and androgen receptor signaling. Cellular signaling pathways altered in *ETV4*-fusion-positive cancer were validated by analyzing microarray data from a previous study [18]. In the study, 774 genes were altered by either overexpressing or knocking-down *ETV4* in a prostate cancer cell line, and signaling pathways were analyzed using these genes. When ORA was performed using 774 genes, metabolism, interferon signaling, and MAPK/NF-kB signaling, among seven inferred pathways, were consistent with the results of this study using the TCGA database (Figure 3).

### 3.4. Identifying Therapeutic Targets and Potential Actionable Drugs

To suggest therapeutic molecular targets and candidate drugs for *ETV4*-fusion-positive prostate cancer, gene–drug network analysis was performed using 86 genes involved in cancer-specific pathways of *ETV4*-fusion-positive prostate cancer. *PARP1*, *NQO1*, *HSPA5*, and *TOP1* were identified as potential molecular targets for *ETV4*-fusion-positive prostate cancer, and olaparib, amrubicin, fluorouracil, and irinotecan were suggested as candidate drugs, respectively (Table 2). *HSPA5* and *TOP1* were associated with the irinotecan pathway and androgen receptor signaling, while *PARP1* was categorized under the androgen receptor signaling pathway, and *NQO1* was related to metabolism (Figure 1).

## 4. Discussion

*ETV4* rearrangement has lately been discovered as a driver gene in 2% of prostate cancers. The small number of *ETV4* subtypes among prostate patients and the short history of research on *ETV4* rearrangement have limited understanding of the molecular features of *ETV4* subtypes of prostate cancer and, in turn, have resulted in an absence of effective targeted therapy for these patients. In addition, since *ETV4* is a transcription factor, it is difficult for it to be a direct drug target, unlike proteins that have a kinase domain or cell-surface receptor [4]. Considering that *ETV4* rearrangement is mutually exclusive with other oncogenic driver mutations in prostate cancer, the study of the molecular features, specific therapeutic targets, and potential drugs for *ETV4* subtypes of prostate cancer are needed. In this study, we first identified 393 genes correlated with *ETV4* expression and six cancer-specific cellular signaling pathways, namely the irinotecan pathway, metabolism, androgen receptor signaling, interferon signaling, MAPK/NF-kB signaling and the tamoxifen pathway, altered by the presence or absence of *ETV4* fusion. These signaling pathways could be further categorized into three groups: hormone-related pathways, metabolic pathways, and inflammation/cancer pathways. 

Androgen signaling is stimulated by androgen, and activated androgen signaling multiplies and spreads prostate cancer cells [19]. Anti-androgen therapy is used to stop the growth of and shrink cancer cells [20,21]. We previously reported that *ERG* subtypes of prostate cancer have altered androgen receptor signaling, similar to the *ETV4* subtypes of prostate cancer shown in the current study [22]. A previous study showed that anti-androgen therapy has a better effect on tumorigenesis in the *ERG* subtypes of prostate cancer [23], and we expect a similar effectiveness from anti-androgen therapy in *ETV4*-fusion-positive prostate cancer, based on its unique features of cellular signaling pathways. Estrogen treatment was once suggested as an alternative method to suppress testosterone in prostate cancer [24], and consistently, we found that the tamoxifen pathway was associated with *ETV4*-fusion-positive prostate cancer, and all genes related to the tamoxifen pathway were positively correlated with *ETV4*. These observations suggest the advantage of using hormonal therapy in *ETV4*-fusion-positive prostate cancer. As half of the genes included in the six cancer-specific pathways were related to the metabolism pathway, interestingly, the metabolic pathway was highly associated with the *ETV4* subtype of prostate cancer (Figure 1). Metabolism has a close connection to cancer. When a normal prostate cell transforms into a prostate cancer cell, many alterations in metabolism occur, including lipid and glucose metabolism [25,26,27]. Also, genes or mechanisms related to the metabolism pathway have been used as therapeutic targets [25,28]. This suggests that our selected genes in the metabolism pathway could be possible therapeutic targets for *ETV4*-fusion-positive cancer. MAPK/NF-kB signaling was also altered in the *ETV4* subtype of prostate cancer (Figure 1). The MAPK and NF-kB pathways are common inflammatory signaling pathways. Activation of the MAPK and NF-kB pathways causes the release of proinflammatory cytokines, such as interleukin-6 (IL-6), IL-8, and tumor necrosis factor-a (TNF-a), which leads to the inflammatory response, and eventually causes cancer. Several studies have shown that inhibition of MAPK or NF-kB pathway activation decreases prostate tumor progression, such as proliferation, invasion, and migration, and increases apoptosis in various cancer types, including prostate cancer [29]. 

Next, we performed gene–drug network analysis and identified new therapeutic targets and drugs for *ETV4*-fusion-positive prostate cancer. Genes associated with *ETV4*-fusion-positive prostate cancers were used as the input, and four therapeutic candidate targets, *PARP1*, *NQO1*, *HSPA5*, and *TOP1*, and the respective selective drugs, olaparib, amrubicin, fluorouracil, and irinotecan, were identified. Olaparib, the first targeted drug for prostate cancer, has recently been approved for *BRCA*-mutated metastatic castration-resistant prostate cancer [30]. In addition, our study suggests that olaparib could also be actionable in the *ETV4* subtype of prostate cancer, regardless of *BRCA* mutation. A large clinical study should be warranted for evaluation of the effects of olaparib on the *ETV4* subtype of prostate cancer in the future. 

Furthermore, we found different gene expression patterns in different types of prostate cancer. For example, *PXN*, which is known as an oncogene in prostate cancer [15], has higher expression in the *ERG* and *SPOP* subtypes of prostate cancer, as compared to normal cells, and lower expression in *ETV1* and *ETV4* subtypes of prostate cancer, as compared to normal cells (Appendix A). This shows that understanding the molecular features of each cancer subtype is a prerequisite for precision medicine. For example, *PXN* is known to be related with tumor progression, invasion, and metastasis via EMT [31,32,33,34], and the expression analysis in this study suggests that a prostate cancer subtype with *ERG* fusion or *SPOP* mutation could share this cancer progression mechanism while a prostate cancer subtype with an *ETV4* fusion or *ETV1* fusion does not.

In this study, cellular signaling pathway analysis was extensively performed, and novel therapeutic targets and candidate drugs for *ETV4*-fusion-positive prostate cancer patients were systematically identified. Cancer genomic data, cancer transcriptomic data, clinical data, and patient information were obtained from the TCGA database. Microarray data performed in cell lines after upregulating or downregulating *ETV4* was collected from prior study data and utilized to conclude the molecular function of *ETV4*. The future validation of the potential therapeutic targets and potential drugs in in vitro or in vivo models would demonstrate the efficiency of our approach.

*ETV4* overexpression was observed in various cancer types, such as prostate cancer, bladder cancer, gastric cancer, colon cancer, and hepatocellular carcinoma, and the underlying mechanism and oncogenic function have been revealed, which suggest the importance of *ETV4* in cancer research [35]. Therefore, our study will help to advance and accelerate the understanding *ETV4*-associated human cancers.

## 5. Conclusions

In this study, we discovered that six cancer-specific cellular signaling pathways, the irinotecan pathway, metabolism, androgen receptor signaling, interferon signaling, MAPK/NF-kB signaling, and the tamoxifen pathway, were altered in the *ETV4* subtype of prostate cancer. To suggest potential therapeutic targets and actionable candidate drugs, gene–drug network analysis was performed, and four genes, *HSPA5*, *NQO1*, *PARP1*, and *TOP1*, and their targeted drugs, fluorouracil, amrubicin, olaparib, and irinotecan, respectively, were identified. 

Taken together, we have provided information on altered cellular signaling pathways and therapeutic targets for *ETV4*-fusion-positive prostate cancer. Considering the absence of targeted drugs for the *ETV4* subtype of prostate cancer, our study will allow a step forward in investigations of treatments. Furthermore, the analysis algorithms we established in this study could be applied to analyze the molecular features of other types of cancer, and in turn, this could provide enormous benefits in developing new drug treatments in regard to cost, time, and the prediction of adverse effects.

## Figures and Tables

**Figure 1 biomedicines-10-02650-f001:**
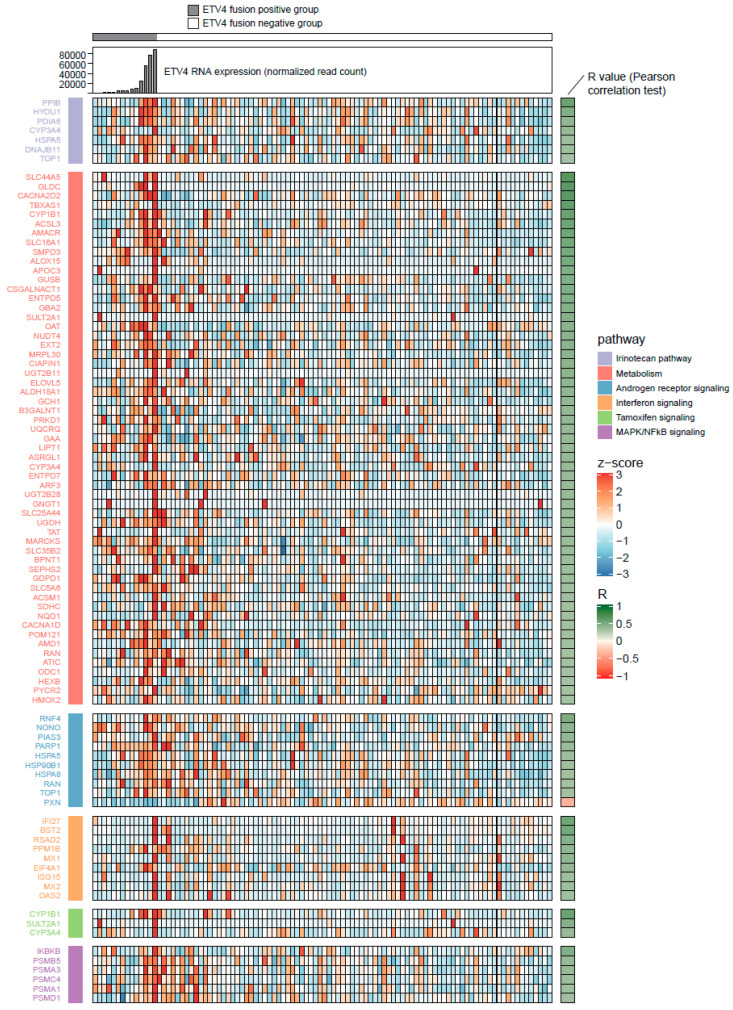
RNA expression heatmap of six cancer-specific pathways. RNA expression was converted into z-scores. Rows include each gene of the cancer-specific pathways arrayed by each pathway. Columns include each sample of an *ETV4*-fusion-positive or -negative group.

**Figure 2 biomedicines-10-02650-f002:**
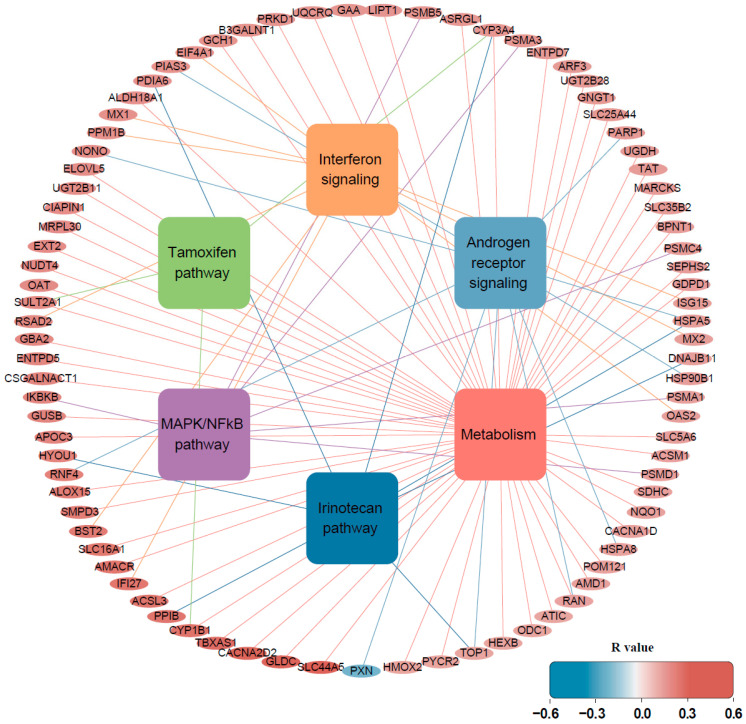
Gene-pathway network. Round rectangles are pathways, and ellipses are genes. The colors of the lines originate from the colors of each rectangle of the pathway.

**Figure 3 biomedicines-10-02650-f003:**
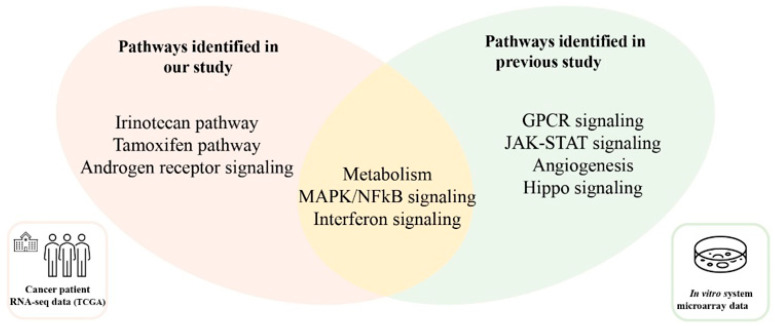
Comparison of pathways found from two studies on *ETV4*-related changes. Pathway analysis was performed by CPDB. This study used 393 genes correlated with *ETV4* expression, and the previous study used 774 genes altered by overexpression or knock-down of *ETV4* [18].

**Table 1 biomedicines-10-02650-t001:** Comparison of patient clinical characteristics of prostate cancer groups.

	*ETV4*-Fusion-Positive (*n* = 14)	*ETV4*-Fusion-Negative (*n* = 86)	*p*-Value
**Age**	60.8 ± 6.3 (*n* = 9)	61.6 ± 6.1 (*n* = 68)	0.653
**Ethnicity**			0.850
Asian	0.0% (0/7)	4.4% (2/45)	
African American	14.3% (1/7)	13.3% (6/45)	
White	85.7% (6/7)	82.2% (37/45)	
**PSA level**	17.5 ± 25.0 (*n* = 14)	14.0 ± 17.0 (*n* = 82)	0.626
**Laterality**			0.703
bilateral	92.9% (13/14)	89.4% (76/85)	
left	0.0% (0/14)	4.7% (4/85)	
right	7.1% (1/14)	5.9% (5/85)	
**Gleason score**			0.398
6	7.1% (1/14)	9.3% (8/86)	
7	64.3% (9/14)	54.7% (47/86)	
8	0.0% (0/14)	11.6% (10/86)	
9	21.4% (3/14)	23.3% (20/86)	
10	7.1% (1/14)	1.2% (1/86)	
**M stage**			0.317
M0	92.9% (13/14)	100.0% (81/81)	
M1b	7.1% (1/14)	0.0% (0/81)	
**T stage**			0.467
T1c	40.0% (4/10)	51.4% (36/70)	
T2	10.0% (1/10)	1.4% (1/70)	
T2a	10.0% (1/10)	5.7% (4/70)	
T2b	10.0% (1/10)	8.6% (6/70)	
T2c	0.0% (0/10)	15.7% (11/70)	
T3a	20.0% (2/10)	11.4% (8/70)	
T3b	10.0% (1/10)	2.9% (2/70)	
T4	0.0% (0/10)	2.9% (2/70)	

**Table 2 biomedicines-10-02650-t002:** Actionable drug list and targeted gene.

Gene	Signaling Pathway	Drug
*TOP1*	Irinotecan pathway Androgen receptor signaling	Irinotecan
		Topotecan, Carboplatin, Cyclophosphamide
*HSPA5*	Irinotecan pathway Androgen receptor signaling	Fluorouracil
*NQO1*	Metabolism	Amrubicin
*PARP1*	Androgen receptor signaling	Olaparib

## Data Availability

Not applicable.

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
