# Peer review of "Portrait of Molecular Signaling and Putative Therapeutic Targets in Prostate Cancer with ETV4 Fusion"

_biomedicines, 2022, doi:10.3390/biomedicines10102650_

Round 1
Reviewer 1 Report
In this manuscript the authors identified significantly altered genes and oncogenic signaling pathways in ETV4 fusion positive prostate cancer and suggested therapeutic targets and potential drugs for ETV4 fusion positive prostate patients. Overall interesting results. The manuscript is well written and presented.
However, I have some comments before acceptance.
In figure 1 I couldn’t find the “Ref” in the figure.
Figure 2 is not clear. Please enlarge it.
The manuscript would benefit from such parallelism with literature
https://doi.org/10.3389/fcell.2021.623809
https://doi.org/10.1016/j.omto.2021.07.012
10.1016/j.eururo.2009.04.036
Reviewer 2 Report
The authors present a very interesting analysis of The Cancer Genome Atlas data, referring to Ets variant transcription factor 4. They selected “sample type code 01, indicating primary solid tumor samples, and sample type code 11, indicating adjacent normal tissue”, so for the 14 ETV4 fusion positive prostate cancers in table 1 (patient clinical characteristics) ¿seven patients were selected (1 african-american and 6 white patients)? Firstly, the term “race” might be considered to be changed, and secondly, the sample size should be validated in methodology; this is very important considering that the conclusions are based on the differences for the “ETV4 positive group”, but in Figure 1, only three or four samples in this group have high ETV4 expression. The authors do not indicate how adjacent normal tissue was analyzed for gene expression. The authors do not discuss either the difference in PSA level, taking into account the great variability in ETV4 positive group, so one or two patients are probably giving the major differences. PXN downregulation in ETV4 fusion positive prostate cancer is a remarkable result. The analysis of the different pathways that are involved is also interesting, although figure 2 does not have an adequate quality to appreciate the gene interactions. “Metabolism” is an ambigous proposal, although it is expected for some genes to be found in more than two pathways, as the authors have commented; glucose metabolism, lipid metabolism, etc could be analyzed. The proposals for precision medicine is well documented, although as the authors have said, they need “validation of the potential therapeutic targets and potential drugs”.
Round 2
Reviewer 2 Report
The authors present a revised version of the manuscript with adequate changes in response to this reviewer suggestions; there is only some clarification needed. For the sufficiency of data from TCGA, with only 7 patients information (from the 14 patients considered for ETV4 fusion positive group), the results for % need revision: 85.7% for 6 ETV4 fusion positive patients and 82.2% for 37 ETV4 fusion negative patients, for example, or precise description in methodology, as it was explained in response cover letter..
